# The Pathway to Cancer Cachexia: MicroRNA-Regulated Networks in Muscle Wasting Based on Integrative Meta-Analysis

**DOI:** 10.3390/ijms20081962

**Published:** 2019-04-22

**Authors:** Paula Paccielli Freire, Geysson Javier Fernandez, Sarah Santiloni Cury, Diogo de Moraes, Jakeline Santos Oliveira, Grasieli de Oliveira, Maeli Dal-Pai-Silva, Patrícia Pintor dos Reis, Robson Francisco Carvalho

**Affiliations:** 1Department of Morphology, Institute of Biosciences, São Paulo State University (UNESP), Botucatu, São Paulo 18.618-619, Brazil; paula.freire@unesp.br (P.P.F.); jasonfergar@hotmail.com (G.J.F.); sarahscury@gmail.com (S.S.C.); demoraesdiogo2017@gmail.com (D.d.M.); jakoliveira.jo@gmail.com (J.S.O.); oliveira.grase@gmail.com (G.d.O.); maeli.dal-pai@unesp.br (M.-D.-P.-S.); 2Department of Surgery and Orthopedics, Faculty of Medicine, São Paulo State University (UNESP), Botucatu, São Paulo 18.618-687, Brazil; patricia.reis@unesp.br; 3Experimental Research Unity, Faculty of Medicine, São Paulo State University (UNESP), Botucatu, São Paulo 18.618-687, Brazil

**Keywords:** cancer cachexia, microRNAs, transcriptome, protein-protein interaction networks

## Abstract

Cancer cachexia is a multifactorial syndrome that leads to significant weight loss. Cachexia affects 50%–80% of cancer patients, depending on the tumor type, and is associated with 20%–40% of cancer patient deaths. Besides the efforts to identify molecular mechanisms of skeletal muscle atrophy—a key feature in cancer cachexia—no effective therapy for the syndrome is currently available. MicroRNAs are regulators of gene expression, with therapeutic potential in several muscle wasting disorders. We performed a meta-analysis of previously published gene expression data to reveal new potential microRNA–mRNA networks associated with muscle atrophy in cancer cachexia. We retrieved 52 differentially expressed genes in nine studies of muscle tissue from patients and rodent models of cancer cachexia. Next, we predicted microRNAs targeting these differentially expressed genes. We also include global microRNA expression data surveyed in atrophying skeletal muscles from previous studies as background information. We identified deregulated genes involved in the regulation of apoptosis, muscle hypertrophy, catabolism, and acute phase response. We further predicted new microRNA–mRNA interactions, such as miR-27a/*Foxo1*, miR-27a/*Mef2c*, miR-27b/*Cxcl12*, miR-27b/*Mef2c*, miR-140/*Cxcl12*, miR-199a/*Cav1*, and miR-199a/*Junb*, which may contribute to muscle wasting in cancer cachexia. Finally, we found drugs targeting *MSTN*, *CXCL12*, and *CAMK2B*, which may be considered for the development of novel therapeutic strategies for cancer cachexia. Our study has broadened the knowledge of microRNA-regulated networks that are likely associated with muscle atrophy in cancer cachexia, pointing to their involvement as potential targets for novel therapeutic strategies.

## 1. Introduction

Cachexia is a syndrome associated with pathological conditions, including sepsis, chronic obstructive pulmonary disease, heart failure, and cancer [1,2]. Notably, cachexia is the leading cause of death for 20%–40% of cancer patients [3], and affects around 60% of patients when all cancer types are considered [4]. Cachexia is more prevalent in gastric or pancreatic cancer, as up to 80% of patients may develop the syndrome [5,6]. 

Cachexia occurs in all cancer stages, and is associated with poor prognosis, decreased treatment tolerance, and a significant reduction in quality of life [7]. International consensus defines the diagnostic for cancer cachexia based on weight loss greater than 5% over six months, or weight loss greater than 2% in individuals with a body mass index lower than 20 kg/m^2^ or with sarcopenia [7]. Other features associated with cancer cachexia are a reduction in food intake, an increase of systemic inflammation markers like C-reactive protein, and decreased response to chemotherapy [7]. These features may increase surgical risk in cachectic patients [7,8]. Thus, the conservation of lean body mass is critical for cancer patients’ survival. Although there are some advances in therapeutic strategies for muscle wasting in cancer cachexia (reviewed in [4]), effective targets to treat the syndrome are still lacking. 

Many studies have been conducted to identify the molecular mechanisms related to muscle wasting in cancer cachexia. These studies have already lead to important advances through the recognition of the association between cachexia and high levels of pro-inflammatory cytokines, such as interleukin (IL)-1β, IL-6, IL-8, tumor necrosis factor alpha (TNF), and interferon gamma (IFN) [9,10,11,12,13,14]. These cytokines activate different molecular axes in skeletal muscle cells by the nuclear factor kappa-light-chain-enhancer of activated B cells (NF-κB), signal transducer and activator of transcription (STAT), MAP kinase family (MAPKs), and activator protein 1 (AP1) [15,16,17,18]. Signal transductions to NF-κB and STAT transcription factors have key roles, especially in altering three major effector biological systems: the ubiquitin–proteasome system, the IGF1–AKT–FOXO signaling pathway, and the autophagy–lysosome system [19,20,21]. Together, these systems contribute to an imbalance between protein synthesis and degradation that results in loss of muscle mass and function [19,20,21]. Given the complexity of these processes leading to muscle atrophy, the identification and characterization of new genes and signaling pathways based on global analysis will likely contribute to the understanding of the underlying molecular mechanisms of muscle wasting in cancer cachexia. 

In fact, global gene expression studies of muscle wasting conditions, such as glucocorticoid treatment, immobilization, unloading, diabetes, sarcopenia, starvation, and denervation [22,23,24,25,26,27], have helped to shed light on the molecular mechanisms of muscle atrophy, including the identification of new potential biomarkers of cancer cachexia [28,29]. The identification of microRNAs has also broadened the knowledge about global gene expression regulation in conditions that induce skeletal muscle atrophy [30,31,32,33], including cancer cachexia [34,35]. However, there is a lack of integration of global microRNA and mRNA expression data from the same set of muscle samples in previous cancer cachexia studies [28,29,34,35,36,37,38,39,40,41]. Furthermore, to our knowledge, no study has integrated the most relevant microRNA and mRNA data available in the literature. Such integrative strategies are important for identifying the functional significance of key deregulated genes, microRNAs, and molecular pathways involved in muscle wasting in cachexia.

Moreover, the high diversity among human and animal models with cancer cachexia, the complexity of the syndrome, the difficulties in recruiting patients for studies, and the heterogeneity of cancer cells and muscle phenotypes leads to low translatability from experimental systems to clinical practice [42]. We performed a systematic integration of validated gene expression data derived from global mRNA and microRNA expression profiles in muscle wasting in cancer cachexia, to capture the most relevant microRNA-regulated networks across multiple human and rodent studies. Our analysis identified new molecular pathways potentially involved in skeletal muscle atrophy in cancer cachexia. These data have also proven useful for identifying new, potentially molecular-targeted treatment strategies for the syndrome.

## 2. Results 

### 2.1. Study Selection and Characteristics

The meta-analysis resulted in nine studies reporting skeletal muscle gene expression data in cancer cachexia [28,29,36,37,38,39,40,41,43]. The process to select the studies is summarized in Figure 1. We present the description of publicly available cancer cachexia studies included in the meta-analysis in Table 1. These studies report muscle gene expression data from patients and mouse models with different cancer types (gastrointestinal, colon, and pancreatic cancers). The expression data were obtained in distinct muscles: three studies present data from the gastrocnemius, three from the quadriceps, one from the rectus abdominis, one from the biceps femoris, and one from the extensor digitorum longus. This variability in skeletal muscle phenotype expands the transcript profiles, increasing the number of possibilities for identification of key common molecular pathways in muscle wasting triggered by different cancer types. Most of these studies were performed in a limited number of samples (3–21 samples/group), and therefore, a comprehensive and integrative analysis of these data may reveal new molecular components that are not identified when these studies are analyzed individually.

### 2.2. Validated Data Selection of Differentially Expressed Genes in Cancer Cachexia

We filtered the genes with differential expression that were validated by western blotting and/or quantitative reverse transcription polymerase chain reaction (RT-qPCR) techniques. This strategy allowed us to use the transcripts selected as more relevant by the authors in the global analysis, and therefore, specifically investigate those that may be directly related to molecular alterations in muscle wasting in cancer cachexia. These studies reported, excluding duplicates, 52 differentially expressed genes in 59 samples of muscle tissue from patients and rodent models of cancer cachexia. A list of the validated data for differentially expressed genes in cancer cachexia, with their respective functions and location, is summarized in Table 2 and Appendix A. We highlight that the atrogenes *Fbxo32* and *Trim63* appeared in six out of the nine selected studies, and *Cebpd* and *Cxcl12* are dysregulated in two studies. Notably, 10 over-expressed genes (*Comp, Mmp3, Adipoq, Angptl7, Fgg, Hp, Mstn, Saa1, Serpina3n,* and *Cxcl12*) are translated into secreted proteins, and therefore, can be further explored as potential cancer cachexia biomarkers. 

### 2.3. Gene Ontology Enrichment Analysis of Differentially Expressed Genes in Muscle Wasting in Cancer Cachexia

Gene ontology (GO) analysis shows information on the biological role of differentially expressed genes involved in muscle wasting in cancer cachexia. We used gene ontology hierarchically structured categories to identify proteins encoded by the up- and down-expressed genes. This analysis revealed over-represented GO categories of biological processes that included structural and development genes (e.g., negative regulation of muscle hypertrophy, anatomical structure morphogenesis, epithelial cell proliferation, muscle organ development, muscle cell differentiation, and tissue development), metabolic process, acute-phase response, and apoptotic process. Other relevant terms enriched in our dataset included response to insulin and response to hormone stimulus. The statistically significant enriched GO terms are shown in Figure 2.

### 2.4. Protein–Protein Interaction Network in Muscle Wasting in Cancer Cachexia

The integrated protein–protein interaction (PPI) network shows a higher number of interactions between proteins of the inflammatory response, catabolism and anabolism, fat metabolism, apoptotic process, and transcriptional control. Complex interactome analysis of deregulated genes in cancer cachexia, with respective functional annotations, is illustrated in Figure 3.

### 2.5. Identification of New Potential microRNA-Regulated Networks in Muscle Wasting in Cancer Cachexia

The miRNA-mRNA target prediction identified 3150 non-validated and 98 validated interactions (Appendix A). The validated interactions were used to construct a microRNA-target mRNA interaction network for up- and down-regulated genes (Figure 4A,B, respectively). Networks were constructed with our list of microRNAs predicted in silico as targeting the mRNAs retrieved by our meta-analysis, and with microRNAs found as deregulated in two previous microRNAs studies on muscle wasting in cancer cachexia [34,35] (Appendix A). Interestingly, the intersection of all the microRNA data showed that our list of predicted microRNAs shares five microRNAs with one study [35] (miR-27a, miR-27b, miR-140, miR-24, and miR-15) and the microRNA miR-199 with another [34] (Figure 5A). 

Next, we predicted genes targeted by the differentially expressed microRNAs by both these previous studies. These genes were further compared with the list of 52 deregulated genes identified in our meta-analysis. We found a total of five shared transcripts (*Cav1*, *Cxcl12*, *Foxo1*, *Mef2c*, and *Junb*) (Figure 5B), and most importantly, these five transcripts revealed seven new microRNA-mRNA interactions in muscle wasting in cancer cachexia (miR-27a/*Foxo1*, miR-27a/*Mef2c*, miR-27b/*Cxcl12*, miR-27b/*Mef2c*, miR-140/*Cxcl12*, miR-199a/*Cav1*, and miR-199a/*Junb*) (Table 3). Notably, three interactions—miR-27a/*Foxo1*, miR-140/*Cxcl12*, and miR-199a/*Cav1*—showed an opposite direction of expression between microRNAs (identified in the previous studies [34,35]) and mRNAs (identified in the meta-analysis) (Table 3).

### 2.6. Identification of Potential Target Agents for the Treatment of Muscle Wasting in Cancer Cachexia

Interestingly, Drug–Gene Interaction Database (DGIdb) data revealed *ADIPOQ*, *CAMK2B*, *COMP*, *CXCL12*, and *MSTN* as drug-targetable genes, using chemical compounds such as Spironolactone, Fenofibrate, Nevirapine, Mibefradil, Nifedipine, Nisoldipine, Tadalafil, Tinzaparin Sodium, and Stamulumab (Table 4). The above drugs have been demonstrated as clinically useful in Type 2 diabetes mellitus, cardiovascular disease, HIV infection, smooth muscle, cardiac muscle, breast cancer, and myopathies [44,45,46,47,48,49,50,51,52,53]. Outstandingly, the potential drugs found here have not been tested yet for the treatment of muscle wasting in cancer cachexia.

## 3. Discussion

Researchers have conducted several studies on molecular mechanisms of muscle wasting in cancer cachexia. However, the complexity of the syndrome and the insufficient knowledge of pathogenic mechanisms hinder the design of effective therapeutic strategies. Most cancer cachexia studies rely on a single, thorough, standardized model or specific cancer types, rarely integrating and comparing their datasets with those from other experimental systems. The present meta-analysis allowed us to select relevant mRNA expression data from different cancer cachexia studies. This is the first study integrating the most relevant literature data from global gene expression profiling studies in muscle wasting in cancer cachexia, in order to find common regulatory networks and molecular pathways. We identify new potential microRNA-regulated gene networks involved in muscle wasting in cancer cachexia (Scheme 1). Specifically, our results suggest that microRNA/mRNA interactions miR-27a/*Foxo1*, miR-27a/*Mef2c*, miR-27b/*Cxcl12*, miR-27b/*Mef2c*, miR-140/*Cxcl12*, miR-199a/*Cav1*, and miR-199a/*Junb* may contribute to muscle wasting in cancer cachexia.

Among the 52 deregulated genes identified in our analysis, six of the nine studies included in the meta-analysis have evaluated the expression of *Trim63* and *Fbxo32* as molecular markers of muscle atrophy in cancer cachexia. In 2001, *Trim63* and *Fbxo32* were first identified as two important muscle-specific E3 ubiquitin ligases that are transcriptionally increased in skeletal muscle under atrophy-inducing conditions, making them excellent markers of muscle atrophy [54]. The transcription of enzymes *Trim63* and *Fbxo32* is dependent on transcriptional factors *FoxO1* and *FoxO*3, which are thought to regulate both the ubiquitin/proteasome [27] and autophagy [20,55,56] pathways. Interestingly, our enrichment analysis also showed alteration in the regulation of the apoptotic process induced by changes in the expression of *FoxO1*, *BNIP3,* and *GABARAPL1*. These results are in agreement with the fact that skeletal muscle wasting is the result of an imbalance between synthesis and degradation of protein pathways, together with the instability of regenerative capacity and myocyte apoptosis [57,58]. 

Moreover, we showed miR-145a as a new potential *FoxO1* regulator during muscle wasting in cancer cachexia. Indeed, the increase of miR-145 decreases *FoxO1* expression in metastatic T24T bladder cancer cells [59]. Conversely, in non-metastatic bladder transitional cell carcinoma T24 cells, miR-145 overexpression inhibited cell growth, which correlates with upregulation of *FoxO1* [59]. These opposite results are probably associated with different cell types and experimental conditions, raising the necessity to further explore miR-145-*FoxO1*. One of the most important findings of our study was that *FoxO1* is a validated target of the microRNA miR-27a. Besides, Soares et al. [35] identified the down-regulation of miR-27a in cancer-cachexia. Although the miR-27a-–*FoxO1* interaction is not validated in skeletal muscle cells, the overexpression of miR-27a in mice with chronic kidney disease attenuated muscle loss, improved grip strength, and decreased the expression of FoxO1, Trim63, and Fbxo32 proteins [60]. 

Besides these molecular markers of muscle atrophy, our meta-analysis also identified *Nr3c1* as down-regulated in the skeletal muscle of cachectic patients with upper gastrointestinal cancer [29]. *Nr3c1* has a function in the regulation of muscle hypertrophy and strength in response to resistance training [61]; however, the role of *Nr3c1* in the development of muscle wasting associated with cancer is still unknown. Furthermore, our microRNA target prediction analysis identified nine microRNAs that potentially modulate *Nr3c1* expression, including microRNAs miR-28b and miR-28c. We also identified that the *Nr3c1* transcript is potentially regulated by miR-30e, which is upregulated in the skeletal muscle of *Mstn*-/- mice, and its expression was associated with glycolytic myofiber formation [62]. Considering that microRNAs miR-28b, miR-28c, and miR-30e potentially target the *Nr3c1* transcript, and that both microRNAs and their target transcripts have important functions previously described in skeletal muscle tissue, our integrative analysis reveals these new microRNA–mRNA interactions as potential targets for future exploratory analysis of muscle wasting in cancer cachexia. 

Our prediction analyses also revealed microRNA miR-17 as an important regulator of transcripts, such as *Cxcl12*, *Mef2c*, *Stat3*, and *Cav1*, that are translated into proteins with a role in cancer cachexia. MicroRNA miR-17 is involved in oncogenic events in different cancer types with a high incidence of cachexia (hepatocellular carcinoma [63], pancreatic cancer [64], and non-small lung cancer [65,66]). Among the miR-17 target genes, *Mef2c* is involved in the regulation of skeletal muscle regeneration and myogenesis (reviewed in Dong et al. [67]). Also, two studies have identified miR-27b as a regulator of *Mef2c* [68,69]; notably, one of these studies shows that miR-27b is involved in the regulation of mitochondrial biogenesis in myocytes by regulating *Mef2c* [69]. Importantly, miR-27b also negatively regulates myostatin (*Mstn*) to promote satellite cell activation and myoblast proliferation, and to prevent muscle wasting [70]. *Mstn*, a member of the *TGFβ* superfamily of growth factors, is a highly conserved negative regulator of skeletal muscle mass upregulated in muscle wasting conditions, including cancer cachexia [37,71,72]. Accordingly, *Mstn* deficiency increase skeletal muscle mass and strength and counterattacks muscle wasting conditions [73]. Several studies have demonstrated the therapeutic potential of *Mstn* inhibition under muscle wasting conditions as cancer [74,75]; consequently, the identification of *Mstn* as a putative target of miR-27b has potential therapeutic and biological implications in muscle wasting conditions. 

Moreover, our results showed that the drug Stamulumab (MYO-029), known as a potential *MSTN* inhibitor, was previously tested in clinical studies of subjects with myopathies. This trial showed a potential increase in the muscle size of the subjects, but the researchers observed no improvements in muscle strength or function [52,53]. Considering that there are no studies testing MYO-029 on cancer cachexia, further studies are needed to demonstrate whether this drug may improve muscle mass and function in this syndrome. 

It is also important to highlight that our enrichment analysis identified genes associated with the acute-phase response. Systemic inflammation is a hallmark of cancer cachexia, and this inflammatory response is the main driving force that leads to the metabolic alterations observed in cancer patients [7]. The literature points out several origins of inflammation, including tumor cells and activated immune cells that release cytokines, chemokines, and other inflammatory mediators [76]. *Cxcl12* was one of the top transcripts, with the highest number of microRNA interactions identified in our meta-analysis (nine interactions in total). The identification of microRNAs that target *CXCL12* is important, because it has been demonstrated that *CXCR4* pathway is consistently downregulated in skeletal muscles from mice and patients with cancer-associated cachexia, and the activation of the *Cxcl12/Cxcr4* pathway protects muscle from wasting in mice with the syndrome [43]. This inflammatory chemokine is also relevant in skeletal muscle regeneration by increasing the activity of metalloproteases, which are crucial to the remodeling of the extracellular matrix [77]. Our results also show protein–protein interactions of *Cxcl12* with the metalloprotease *Mmp*, suggesting that this interaction could affect muscle regeneration and extracellular matrix remodeling in cancer cachexia. Indeed, we identified that miR-27b and miR-140 are two potential microRNAs involved in *Cxcl12* regulation. Importantly, it has been previously demonstrated that miR-140 transfection decreases *Cxcl12* expression and release in human airway smooth muscle cells, with a reduction in inflammatory response [78]. Thus, our data also suggest a response that compromises inflammatory response through the mir-140/*Cxcl12* axis in muscle wasting during cancer cachexia.

Furthermore, among the acute phase response genes identified by ontology analysis, *Stat3* was the most notorious factor of our interaction network. The role of this transcriptional factor is widely studied in cancer cachexia. The increase of interleukin-6 in cachectic patients triggers the activation of JAK (Janus kinase), with consequent STAT3 phosphorylation that acts at the nucleus; this leads to transcription activation of several genes associated with skeletal muscle cells growth, atrophy, proliferation, differentiation, survival, and apoptosis [38,79,80]. Moreover, STAT3 contributes to cancer cachexia enhancing tumorigenesis, metastasis, and immune suppression, mostly in tumors associated with a high prevalence of cachexia [80]. Given the diversity of activated genes obtained through the JAK/STAT pathway, it is essential to better characterize *Stat3* downstream target genes in the skeletal muscle cells in wasting conditions. The SOCS3 is a classic inhibitor of the JAK/STAT pathway and several cytokines, and pathogenic mediators induce the expression of SOCS3, which acts in a negative feedback loop to further inhibit signal transduction [81,82]. In a murine model of pancreatic cancer cachexia, the JAK/STAT/SOCS3-dependent intracellular pathway plays an essential role in pathogenesis, since its pharmacological inhibition attenuates cachexia progression in a lethal pancreatic cancer model [39].

Our predicted molecular network also revealed a *Stat3/Junb* interaction. *Junb* is a transcriptional factor that regulates gene expression on multiple levels [83], but the functionality of this *Stat3/Junb* interaction deserves future study in relation to cancer cachexia. Our study identified miR-199a as a potential microRNA that modulates *Junb* expression in muscle wasting in this syndrome. In addition, we found an inverse correlation between miR-199a and *Cav1* expression. Since many inflammatory mediators are activated in cancer cachexia, and the miR199a/*Cav1* axis was previously described in several chronic inflammatory lung diseases as an important regulatory pathway [84], this axis should also be considered for further investigations in muscle wasting in cancer cachexia.

The main contribution of the present investigation is that we identify new potential microRNA-regulated mRNAs in cancer cachexia. Nevertheless, our study has some limitations due to the nature of our analysis, which consists of the reuse of transcriptomic data from different studies and in silico analysis. Further studies are needed to validate the microRNA–mRNA interactions described herein, as well as to validate the efficiency of the identified potential drugs. Furthermore, the deregulated genes selected in our analysis were restricted to those further validated by RT-qPCR or Western Blot. We considered this strategy to increase the possibility of rescuing truly deregulated targets, or with a potential impact on protein levels. Finally, due to the small number of studies evaluating muscle transcriptome of patients with cachexia, we used data from humans and mice together to identify a higher range of transcripts.

In conclusion, we identified new microRNA–mRNA interactions, such as miR-27a/*Foxo1*, miR-27a/*Mef2c*, miR-27b/*Cxcl12*, miR-27b/*Mef2c*, miR-140/*Cxcl12*, miR-199a/*Cav1*, and miR-199a/*Junb*, that may contribute to muscle wasting in cancer cachexia. Finally, we found drugs targeting *MSTN*, *CXCL12*, and *CAMK2B*, which may be considered for the development of novel therapeutic strategies for cancer-related cachexia.

## 4. Methods

### 4.1. Meta-Analysis of Global Gene Expression Data in Muscle Wasting in Cancer Cachexia

We performed a meta-analysis design following the stages of the PRISMA Statement [85] (Figure 1), by searching PubMed (http://www.ncbi.nlm.nih.gov/pubmed) to find the collection of previously published gene expression data of skeletal muscles in cancer cachexia. The keywords used were: “cancer cachexia AND global gene expression”, “cancer cachexia AND transcriptome”, “cancer cachexia AND transcriptomics”, “cancer cachexia AND microarray”, and “cancer cachexia AND RNAseq”. These meta-analysis searches comprised studies published between January 2005 and February 2019. Our inclusion criteria were (1) gene expression data in muscle samples of patients with cancer cachexia or animal models of cancer cachexia, (2) all types of cancer were considered, (3) all types of muscle were considered, (4) the inclusion of normal tissues for comparison, (5) all gene expression analysis platforms were considered, and (6) only data further validated by RT-qPCR or Western Blot were included for the integrative analyses. Our exclusion criteria were (1) non-muscle samples, (2) treatment before molecular genetic analysis, and (3) review studies. The deregulated genes reported in selected studies were further used for bioinformatics prediction of microRNAs as potential regulators of gene expression, as described below.

### 4.2. Meta-Analysis of Global microRNA Expression Data in Muscle Wasting in Cancer Cachexia

To search previously published global microRNA expression data for skeletal muscle in cancer cachexia, we used the following keywords in PubMed: “cancer cachexia AND global microRNA expression”, “cancer cachexia AND microRNome”, and “cancer cachexia AND microRNA profiling”. This search only retrieved four studies: (1) microRNA profiling in adipocyte lipolysis [86]; (2) integrative microRNAs and mRNAs expression analysis during skeletal muscle wasting in cardiac cachexia [87]; (3) microRNA profiling in muscle wasting during catabolic conditions, including cancer cachexia [35]; and (4) microRNA profiling from human skeletal muscle in cancer cachexia [34]. The microRNA data of these last two studies were included for the identification of regulatory networks, in addition to the microRNAs that were identified in an in silico, mRNA-based target prediction described subsequently.

### 4.3. Identification of Muscle microRNAs as Potential Modulators of Deregulated Genes in Cancer Cachexia

The deregulated genes identified in our meta-analysis were used for microRNA prediction by multiple algorithms (TargetScan [88], MiRTarBase [89], and miRWalk [90]) to identify potential regulators (predicted and validated interactions) of the expressed genes in cancer cachexia. Next, to generate the interaction networks, we filtered all microRNAs found by these computational tools, considering only those identified by MiRTarBase as presenting the “reporter assay” as a validation method. We selected MiRTarBase [89] due to its different validation methods of interaction between mRNAs and microRNAs, ranging from strong to weak evidence of interaction (accordingly, “reporter assay” has the strongest evidence of microRNA–target gene interaction). We also used MiRTarBase to identify microRNA-target transcripts from global microRNA expression in cancer cachexia studies [34,35]. The deregulated genes were used to identify over-represented gene ontology categories of biological processes with the Gene Ontology Consortium tool, powered by PANTHER v11.0 [91,92,93] (available at http://www.pantherdb.org/). We considered the GO categories with *p*-value ≤ 0.05 to be significant. The UniProtKB database (available at http://www.uniprot.org/) was used to access functional information of components identified through the meta-analysis. Protein–protein interaction (PPI) networks were then generated using Metasearch STRING v10.5.1 [94,95]. Visualization and annotation data of PPI and microRNA-gene interaction networks were generated using Cytoscape v3.4.0 [96].

### 4.4. Identification of Candidate Drug Targets Based on microRNA-Regulated Networks in Cancer Cachexia

We used the Drug–Gene Interaction Database (DGIdb), a database and web interface for finding known and potential drug–gene relationships. Genes were defined by Entrez Gene and Ensembl and were matched with genes from drug–gene interactions and druggable gene categories. The drugs were defined by searching PubChem, and were then matched with drugs from drug–gene interaction data. The source guide to pharmacology interaction was obtained from the DrugBank [97,98].

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
