# Peer review of "The Pathway to Cancer Cachexia: MicroRNA-Regulated Networks in Muscle Wasting Based on Integrative Meta-Analysis"

_ijms, 2019, doi:10.3390/ijms20081962_

Round 1

Reviewer 1 Report

 The study provides insight into novel signalling pathways that are potential novel targets for therapy of cachexia as a systemic syndrome. Especially, the finding that CXCL12 and systemic inflammation are associated to cachexia is crucial. However, mechanisms that might be involved in metabolic driven regulation of  adaptive and innate immune fate decision and funtion by the CXCL12 and CXCR4 axis  and how these might mediate cachexia progression should be discussed. The fact that inflammation is a risk factor for cachexia is crucial to understand the syndrome in many chronic inflammatory and metabolic diseases and to identify novel targets for prevention of cachexia in patients wit these diseases.

Author Response

Response 1:

Thank you for the comments on our manuscript. We have now included in our discussion the required information regarding the CXCL12 and CXCR4 axis in muscle wasting during cancer cachexia, as demonstrated by Martinelli et al., 2016 [1] .

Reference

Martinelli, G.B.; Olivari, D.; Re Cecconi, A.D.; Talamini, L.; Ottoboni, L.; Lecker, S.H.; Stretch, C.; Baracos, V.E.; Bathe, O.F.; Resovi, A.; et al. Activation of the SDF1/CXCR4 pathway retards muscle atrophy during cancer cachexia. Oncogene 2016, 35, 6212–6222.

Reviewer 2 Report

This is an interesting manuscript on cancer cachexia.

It provides relevant information about potential therapeutic targets that required additional studies.

I consider it acceptable for publication in this format.

Author Response

Dear reviewer, 

We appreciate the comments on our manuscript. 
